# Ubericin K, a New Pore-Forming Bacteriocin Targeting mannose-PTS

Thomas F. Oftedal,[a] Kirill V. Ovchinnikov,[a] Kai A. Hestad,[a] Oliver Goldbeck,[b] Davide Porcellato,[a] Judith Narvhus,[a] Christian U. Riedel,[b] Morten Kjos,[a] Dzung B. Diep[a]

[a]Faculty of Chemistry, Biotechnology and Food Science, Norwegian University of Life Sciences, Ås, Norway
[b]Institute of Microbiology and Biotechnology, Ulm University, Ulm, Germany

**ABSTRACT** Bovine mastitis infection in dairy cattle is a significant economic burden for the dairy industry globally. To reduce the use of antibiotics in treatment of clinical mastitis, new alternative treatment options are needed. Antimicrobial peptides from bacteria, also known as bacteriocins, are potential alternatives for combating mastitis pathogens. In search of novel bacteriocins against mastitis pathogens, we screened samples of Norwegian bovine raw milk and found a *Streptococcus uberis* strain with potent antimicrobial activity toward *Enterococcus*, *Streptococcus*, *Listeria*, and *Lactococcus*. Whole-genome sequencing of the strain revealed a multibacteriocin gene cluster encoding one class IIb bacteriocin, two class IId bacteriocins, in addition to a three-component regulatory system and a dedicated ABC transporter. Isolation and purification of the antimicrobial activity from culture supernatants resulted in the detection of a 6.3-kDa mass peak by matrix-assisted laser desorption ionization–time of flight (MALDI-TOF) mass spectrometry, a mass corresponding to the predicted size of one of the class IId bacteriocins. The identification of this bacteriocin, called ubericin K, was further confirmed by *in vitro* protein synthesis, which showed the same inhibitory spectrum as the purified antimicrobial compound. Ubericin K shows highest sequence similarity to the class IId bacteriocins bovicin 255, lactococcin A, and garvieacin Q. We found that ubericin K uses the sugar transporter mannose phosphotransferase (PTS) as a target receptor. Further, by using the pHlourin sensor system to detect intracellular pH changes due to leakage across the membrane, ubericin K was shown to be a pore former, killing target cells by membrane disruption.

**IMPORTANCE** Bacterial infections in dairy cows are a major burden to farmers worldwide because infected cows require expensive treatments and produce less milk. Today, infected cows are treated with antibiotics, a practice that is becoming less effective due to antibiotic resistance. Compounds other than antibiotics also exist that kill bacteria causing infections in cows; these compounds, known as bacteriocins, are natural products produced by other bacteria in the environment. In this work, we discover a new bacteriocin that we call ubericin K, which kills several species of bacteria known to cause infections in dairy cows. We also use *in vitro* synthesis as a novel method for rapidly characterizing bacteriocins directly from genomic data, which could be useful for other researchers. We believe that ubericin K and the methods described in this work will aid in the transition away from antibiotics in the dairy industry.

**KEYWORDS** bacteriocin, ubericin, mastitis, *in vitro* translation, pore formation, quorum sensing, pHluorin

Address correspondence to Dzung B. Diep, dzung.diep@nmbu.no.

Ubericin K, a new pore-forming bacteriocin targeting mannose-PTS.

Bovine mastitis is the most common infection in dairy cattle worldwide and is a major cause of economic losses for the dairy industry due to reduced milk production and quality as well as increased drug and veterinary costs (1). Organisms implicated in bovine mastitis include several species within the genera *Streptococcus*, *Enterococcus*,

and *Staphylococcus* (2, 3). One of the main treatment strategies for bovine clinical mastitis is the use of antibiotics, which is increasingly undesirable due to rising antibiotic resistance and waning efficacy (4). Alternative strategies and agents against mastitis pathogens are therefore needed. A potential alternative to antibiotics for treating bovine mastitis is the use of antimicrobial peptides such as bacteriocins.

Bacteriocins are ribosomally synthesized antimicrobial peptides produced by a broad range of organisms for defense or niche competition, typically targeting closely related species (5). For Gram-positive bacteria, at least two main classes of bacteriocins have recently been established, the posttranslationally modified bacteriocins (class I) and the unmodified bacteriocins (class II) (6–8). Class II bacteriocins are further subdivided into several subclasses. Class IIa consists of the pediocin-like bacteriocins, which contain a conserved YGNG(VL) motif (pediocin box) located near the N-terminal end. These bacteriocins show strong antilisterial activity and kill sensitive cells by membrane disruption and loss of the proton gradient across the membrane (9). Class IIb consists of the two-peptide bacteriocins whose activity requires the presence of two different peptides, normally in equimolar concentrations (10). Class IIc bacteriocins are leaderless and are produced as active peptides without a leader sequence (6). Finally, bacteriocins that are linear and missing the sequence motif characteristic of pediocins are designated class IId (11). A subfamily within class IId is bacteriocins sharing sequence similarity, such as lactococcin A, bacteriocin SJ, garvieacin Q, and bovicin 255 (12–15). These bacteriocins are all translated with a double glycine-type leader sequence and are located near a gene encoding an ABC transporter and peptidase. Maturation of the bacteriocin prepeptide occurs by cleavage at the GG motif (positions −1 and −2), which is coupled to the export of the peptide out of the cell (16).

Unlike lantibiotics, such as nisin, which use the cell wall precursor lipid II as a docking molecule on target cells, the class II bacteriocins appear to use different membrane-located proteins as target receptors. One of them is the sugar transporter mannose phosphotransferase system (man-PTS), which is used as a receptor for most, if not all, class IIa bacteriocins and for some class IId bacteriocins, such as lactococcin A, bacteriocin SJ, and garvieacin Q (17–19). Bacteriocin producers are immune to the action of their own bacteriocins due to immunity proteins that are generally cotranscribed with the bacteriocin gene (17, 20). Immunity proteins are small (50 to 150 amino acids) and are believed to protect the producer by forming a strong complex with the receptor protein and bacteriocin (17).

Lactococcin A kills sensitive cells by forming pores in the cytoplasmic membrane in a manner that depends on the presence of the man-PTS receptor protein (17). Pore formation results in depolarization of the membrane potential of sensitive cells, inhibition of amino acid uptake, and efflux of amino acids already imported (21). This efflux of amino acids is independent of the membrane potential and occurs with membrane vesicles of sensitive cells but not liposomes prepared from phospholipids of sensitive cells (21). This mode of action results in a very potent antimicrobial, as lactococcin A is active at picomolar concentrations (12). Similarly, class IIa bacteriocins as well as other class IId bacteriocins have also been shown to target man-PTS on sensitive cells, and this sugar transport system appears to be an attractive target for different antimicrobial agents (17–19, 22, 23). Man-PTS-targeting bacteriocins are generally highly potent and are, as such, an attractive option for combating pathogens. However, it should also be noted that mutants with resistance to such bacteriocins are observed (24).

To search for novel antimicrobials with the potential to fight bovine mastitis pathogens, we screened a selection of bovine raw milk samples shown to have diverse microbial content (25). Here, we report the identification, purification, and characterization of a new class IId bacteriocin from one isolate of *Streptococcus uberis*. The bacteriocin, called ubericin K, shows antimicrobial activity toward many relevant mastitis pathogens in addition to a potent antilisteria activity. Further, we also show that man-PTS is required for the action of this bacteriocin and that it causes depletion of the proton gradient in target cells.

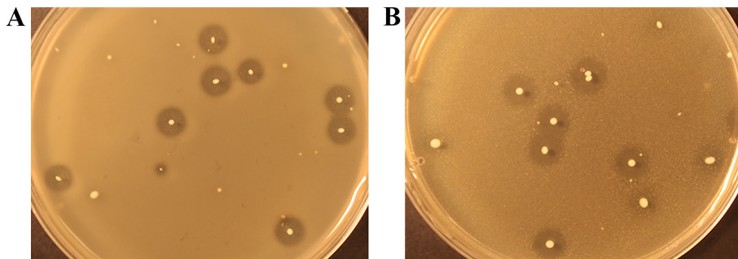

**FIG 1** Screening for bacteriocin producers from sample 385 (10-fold dilution of sample) using *En. faecalis* (A) and *Str. dysgalactiae* (B) as the indicators. Colonies from this sample show clear and defined inhibition zones indicative of bacteriocin production.

## RESULTS

**Bacteriocin screening.** A collection of 53 raw milk samples was screened for antimicrobial producers inhibiting bacteria known to cause various etiological conditions in cows. Only one sample (sample 385) contained several colonies displaying clear inhibition zones against both *En. faecalis* and *Str. dysgalactiae* (Fig. 1). None of the samples had obvious antimicrobial activity against *Sa. aureus*. Sample 385 was obtained from a cow with a high abundance of *Streptococcus* in the udder microbiota, as show in data published previously (25). To determine the identity of the producing colonies and to avoid isolates from the same clone, we performed 16S rRNA gene sequencing and repetitive element PCR of 10 randomly selected colonies with inhibition zones from sample 385. Two unique producers were found, one of *En. faecalis* and one of *Str. uberis* (data not shown). Whole-genome sequencing of both producers was performed to assist identification of the antimicrobials. Using the online bacteriocin prediction tool BAGEL4 (26), the genome of the *En. faecalis* strain was revealed to contain the known enterolysin A gene (27) that encodes enterolysin A, a well-characterized cell wall-degrading protein with a broad inhibitory spectrum that includes *Enterococcus faecalis* (27). The *Str. uberis* strain, however, encoded an uncharacterized bacteriocin-like gene cluster. Due to the novelty of the encoded bacteriocin-like peptides, the *Str. uberis* strain (hereafter called Laboratory of Microbial Gene Technology [LMGT] 4214) was chosen for further study.

**Bacteriocin purification.** Initial physicochemical analyses demonstrated that the antimicrobial activity in the supernatant of LMGT 4214 resisted heating for 10 min at 95°C but was labile to proteinase K treatment (data not shown), properties typical for bacteriocins (28). The antimicrobial activity in the supernatant was purified with a standard bacteriocin purification protocol consisting of three steps: ammonium sulfate precipitation, cation-exchange, and reverse-phase chromatography (29). Purification resulted in a 2,000-fold increase in activity from 40 bacteriocin units (BU)/ml to 81,920 BU/ml with a calculated total yield of 204% (Table 1). A possible explanation for this apparent increase in the amount of bacteriocin is provided in the discussion below. All fractions from reversed-phase fast protein liquid chromatography (RP-FPLC) were assayed for antimicrobial activity against the mastitis pathogen *Str. uberis* LMGT 3912, and only three fractions (10 to 12) were found to have bactericidal activity (Fig. 2). The active fractions eluted in a distinct peak at 31% isopropanol, and the highest activity was found in fraction 11.

**TABLE 1** Purification scheme for ubericin K

| Fraction no. | Fraction | Vol (ml) | Activity (BU/ml) | Total activity (BU) | Yield (%) |
|---|---|---|---|---|---|
| I | Culture supernatant | 1,000 | 40 | 40,000 | 100 |
| II | Ammonium sulphate precipitation | 100 | 320 | 32,000 | 80 |
| III | Cation-exchange chromatography | 100 | 160 | 16,000 | 40 |
| IV | Reversed-phase chromatography | 1 | 81,920 | 81,920 | 204 |

Microbiology
Spectrum

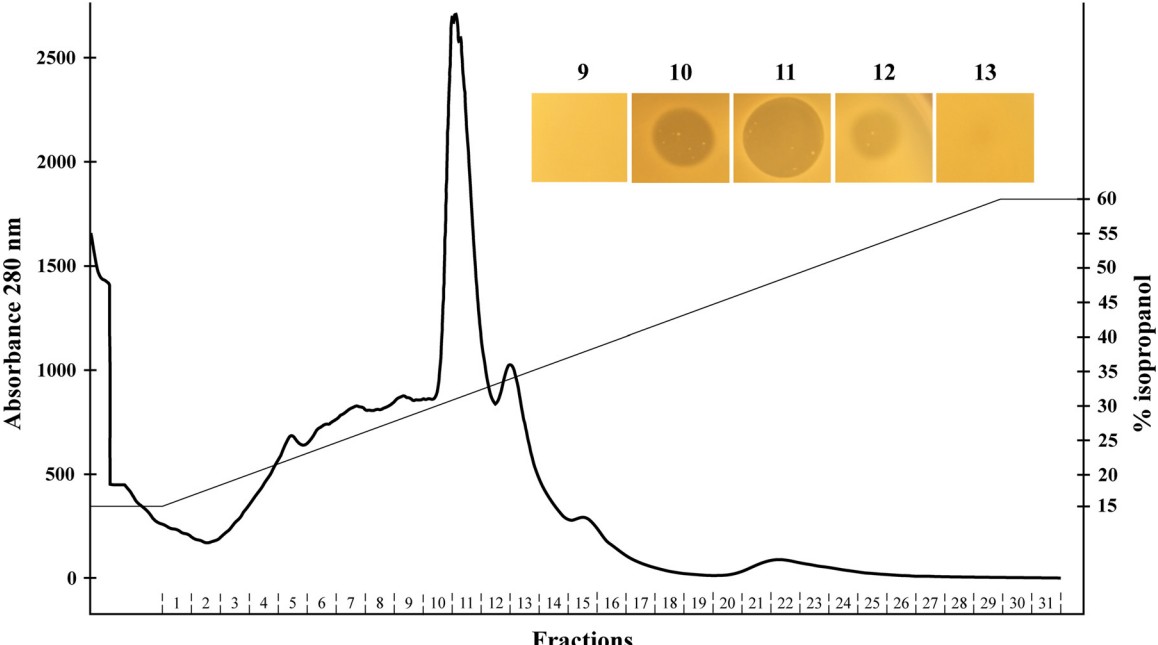

**FIG 2** Reversed-phase chromatography elution profile of crude bacteriocin concentrate obtained from cation-exchange chromatography. Bacteriocin activity against the indicator *Str. uberis* LMGT 3912 was detected in fractions 10 to 12 with the highest activity in fraction 11, which eluted at 32% isopropanol (0.1% [vol/vol] TFA). The inhibition zone produced by a 3-$\mu$l drop of the active fractions is pictured in the upper right.

**Genome sequencing and gene analysis.** Given the proteinaceous nature of the antimicrobial, data from the whole-genome sequencing of LMGT 4214 were used to identify potential bacteriocin genes. Annotation combined with an *in silico* search for bacteriocin genes by the online program BAGEL4 revealed a gene cluster containing bacteriocin-like biosynthetic genes (Fig. 3). A search for homologous DNA in public databases identified several published genomes of *Str. uberis* containing the same locus; however, to our knowledge, genes from this locus have not been studied experimentally. The locus included one open reading frame (ORF) encoding a C39 family bacteriocin-type ABC transporter and peptidase (*orf4*) and several ORFs (*orf3*, *orf6*, *orf7*, *orf10*, and *orf13*) encoding bacteriocin-like peptides, each with a double glycine-type leader sequence (Fig. 3; Table 2). The length of the predicted mature bacteriocin-like peptides varies from 27 amino acids (derived from ORF3) to 58 amino acids (derived

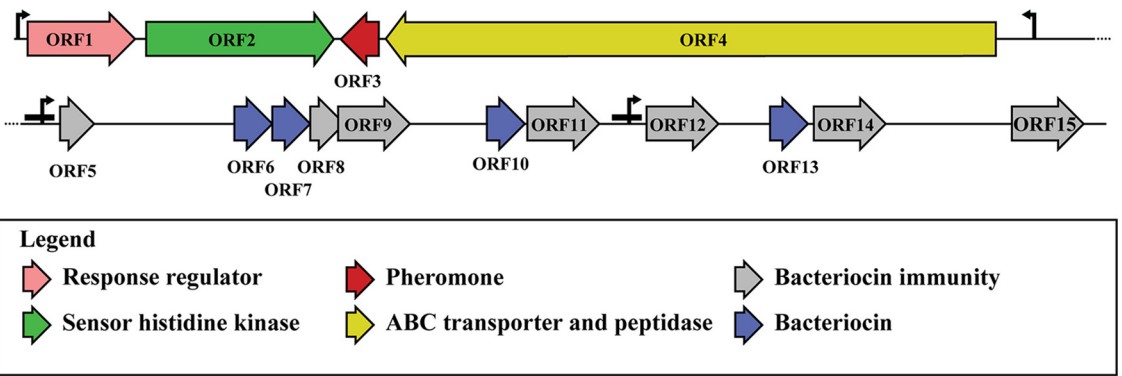

**FIG 3** Genetic organization of the ubericin K bacteriocin cluster. Putative bacteriocins (bacteriocin-like peptides) (blue) and immunity proteins (gray) are located downstream of an ABC transporter and C39 family peptidase (yellow), a putative bacteriocin/pheromone (red), GHKL domain-containing sensor histidine kinase (green), and a LytTR DNA-binding transcriptional regulator (pink). Arrows indicate putative promoters, while the bars upstream of *orf5* and *orf12* indicate potential regulatory repeats. The figure depicts the continuous genomic region from ORF1 to ORF15.

**TABLE 2** List of the encoded proteins from the bacteriocin locus of *Str. uberis* LMGT 4214

| ORF | Predicted function[a,b] | Size (amino acids) | Mass (kDa) | pI | Homologs and/or possible role |
|---|---|---|---|---|---|
| *orf1* | Response regulator | 246 | 29.09 | 9.4 | Response regulator transcription factor (WP_012658029.1) |
| *orf2* | Histidine protein kinase | 439 | 52.47 | 6.7 | Two-component system (TCS) sensor kinase (KKF42577.1) |
| *orf3* | Peptide pheromone | 27 | 3.11 | 12.03 | ComC/BlpC family leader-containing pheromone/bacteriocin (WP_080502297.1) |
| *orf4* | ABC transporter | 717 | 81.51 | 9.38 | Peptide cleavage/export ABC transporter (WP_046392064.1) |
| *orf5* | Unknown | 55 | 6.61 | 9.14 | Hypothetical protein AF69_00955 (KKF59227.1) |
| *orf6* | Bacteriocin-like prepeptide | 53 | 4.99 | 9.31 | Blp family class II bacteriocin (WP_046388584.1) |
| *orf7* | Bacteriocin-like prepeptide | 48 | 4.80 | 9.45 | Blp family class II bacteriocin (WP_046389168.1) |
| *orf8* | Immunity | 70 | 8.46 | 9.52 | Hypothetical protein SAMN05216423_1865 (SEI90296.1) |
| *orf9* | Immunity | 103 | 12.03 | 10.01 | Membrane protein (KKF59107.1) |
| *orf10* | Bacteriocin-like prepeptide | 58 | 6.3 | 9.04 | Garvicin Q family class II bacteriocin (WP_154629194.1) |
| *orf11* | Immunity | 99 | 11.51 | 9.60 | Bacteriocin immunity protein (WP_012658037.1) |
| *orf12* | Unknown | 91 | 10.61 | 9.06 | Hypothetical protein AF68_02745 (KKF60419.1) |
| *orf13* | Bacteriocin-like prepeptide | 54 | 5.87 | 8.68 | Bacteriocin (WP_154590650.1) |
| *orf14* | Immunity | 119 | 14.01 | 9.71 | Bacteriocin immunity protein (MTB58145.1) |
| *orf15* | Unknown | 99 | 11.34 | 9.52 | Bacteriocin immunity protein (WP_154617908.1) |

[a]Predicted function is based on sequence homology, genetic location, or/and physicochemical properties; for immunity proteins, the hydrophobic characteristic, and their genetic location (right after the bacteriocin structural gene) are used for the prediction.
[b]For bacteriocin-like peptides and the pheromone, only their mature sequence was used to calculate size (amino acids), mass (kilodaltons), and pI (isoelectric point).

from ORF10) and all have an alkaline pI (above 8.6). A double glycine leader motif and an alkaline pI are features typical for most unmodified bacteriocins (28, 30). The amino acid sequence of these bacteriocin-like precursors and the expected cleavage site are shown in Fig. 4A.

The bacteriocin-like *orf6* and *orf7* are located next to each other, resembling the genetic organization of a typical two-peptide bacteriocin system (31). These two genes are followed by two small ORFs (*orf8* and *orf9*) encoding small hydrophobic proteins with a probable role in immunity function. The mature sequence of ORF6 and ORF7 share sequence similarities with the class IIb two-peptide bacteriocin lactacin F, 44% and 39% sequence identity with LafA and LafX, respectively (32).

Downstream of *orf9* are *orf10* and *orf11*, which encode a bacteriocin-like peptide and a predicted hydrophobic immunity-like protein, respectively. The predicted mature sequence

**A**

```
                                              ▼
ORF3       MNNFKQFSTLSESELNQIIGG  KSLWRTINSIFNKTTKNAAIKIGNFSR

ORF6       MNSNIEFDSIDTELLEKVIGG  KNNWQANVSGILAAGAAGAAIGAPVCGLACGYIGAKTAITLWAGVTGATGGFK

ORF7        MDKHLVLSNQQLLDVVGG  NAPGNAVLGGLGGLQTGIKYCKVPHPVLKGVCIVGFTTAGAYLAYKAN

ORF10   MKTKSSKQFTELTVKDLSAVIGG  AKGVCKYVYPGSNGYACRYPNGEWGYIVTKSNFEATKDVIVNGWVSSLGGGYFHGNRG

ORF13    MSKMKQFKVLNEELLKKTLGG  TNVAPGIYCVDKNGKAKCSVDYKELWGYTGQVIGNGWINYGPWAPRPGFGVIIP
```

**B**

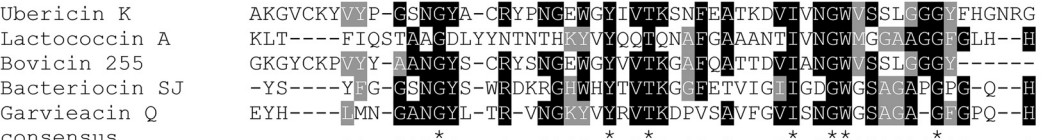

```
Ubericin K      AKGVCKYVYP-GSNGYA-CRYPNGEWGYIVTKSNFEATKDVIVNGWVSSLGGGYFHGNRG
Lactococcin A   KLT----FIQSTAAGDLYYNTNTHKYVYQQTQNAFGAAANTIVNGWMGGAAGGFLH--H
Bovicin 255     GKGYCKPVYY-AANGYS-CRYSNGEWGYVVTKGAFQATTDVIANGWVSSLGGGY------
Bacteriocin SJ  -YS----YEG-GSNGYS-WRDKRGHWHYTVTKGGFETVIGIIGDGWGSAGAFPG-Q--H
Garvieacin Q    EYH----LMN-GANGYL-TR-VNGKVYRVTKDPVSAVFGVISNGWGSAGA-GFGPQ--H
consensus           ..  ...*.  .  ....  *  *.: ..  .  .*  .**......*.. .
```

**FIG 4** List of bacteriocin-like peptides in the ubericin K cluster (A). Predicted leader sequence and mature bacteriocin are separated by a space. The predicted cleavage site is indicated by a triangle (▼). Multiple sequence alignment of known bacteriocins sharing significant sequence identity with ubericin K (ORF10), lactococcin A (M90969.1), bovicin 255 (AF298196), bacteriocin SJ (FM246455), and garvieacin Q (JN605800) (B). The alignment was generated using T-Coffee (http://tcoffee.crg.cat/apps/tcoffee) and colored using BoxShade. Similar amino acids are shaded gray, and identical amino acids are shaded black.

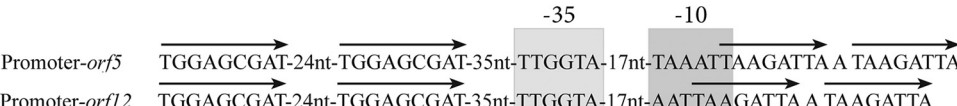

**FIG 5** Putative promoter sequences and potential regulatory elements found upstream of *orf5* and *orf12*. Two direct repeats indicated by the arrows are conserved in the promoter region of both genes.

of ORF10 shows highest similarities to class IId bacteriocins of the garvicin Q family, as shown with lactococcin A, garvieacin Q, and bovicin 255 in Fig. 4B.

Like ORF10, the bacteriocin-like ORF13 is also followed by a small ORF (*orf14*) encoding a small hydrophobic immunity-like protein. The predicted mature part of ORF13 shows sequence homology to hiracin-JM79 (40% identity) (33).

The last bacteriocin-like gene, *orf3*, is located at the other end of the gene cluster between the genes encoding the ABC transporter (*orf4*) and the histidine protein kinase (HPK; *orf2*). Initially, *orf2* was the gene located at the very end of the DNA contig obtained from the Illumina sequencing. HPK genes are often involved in regulation of bacteriocin biosynthesis, where each HPK gene is closely associated with a gene encoding a response regulator (RR) (34). As the HPK gene was located at the very end of this contig, we suspected an RR gene may be in the upstream region of the genome. To prove this, we used the HPK gene as a query to identify nearby genes in sequenced genomes with a similar gene cluster to LMGT 4214. Several homologs to the HPK gene were found. The best hit was the genome of *Str. uberis* NCTC3858 (accession number LS483397.1) in which an annotated RR gene (locus tag NCTC3858_00615) is indeed located next to the searched HPK gene (locus tag NCTC3858_00616). The same RR gene (100% identity) was found in our Illumina sequencing results at the very end of another contig. To confirm that the two contigs were indeed located next to each other in the genome, we performed PCR and Sanger sequencing of the missing region in LMGT 4214. Indeed, using this sequence information, the assembly resulted in a contig with the complete bacteriocin gene cluster (Fig. 3). The RR gene (*orf1*) was located directly upstream of the HPK gene (*orf2*) as expected. The organization of *orf1*, *orf2*, and *orf3* shows a typical genetic organization of a three-component regulatory system containing an HPK (ORF2), an RR (ORF1), and a pheromone peptide (mature product of ORF3), which together are known to be involved in the regulation of biosynthesis in many bacteriocin regulons through a mechanism normally referred to as quorum sensing (35, 36).

An *in silico* search for promoters and regulatory elements was performed. Putative promoters were found just upstream of *orf1*, *orf4*, *orf5*, and *orf12*, suggesting the presence of four operons in the gene cluster (Fig. 3). Interestingly, the two putative promoters situated upstream of *orf5* and *orf12* show only limited homology to consensus promoter sequences, but both are preceded by a pair of direct repeats (TGGAGCGAT) 35 nucleotides (nt) upstream of the −35 box (Fig. 5). The two direct repeats are separated by a spacer sequence of 24 nt. Thus, the distance between the middle of each repeat is 32 to 33 nt, a length corresponding to about three complete helical turns (10.5 bp/turn in B-DNA double helix). This arrangement would allow the two repeats to face toward the same side of DNA, hence resembling regulatory DNA elements for activator binding. Furthermore, downstream of and partially overlapping the −10 box of the two predicted promoters are another pair of direct repeats (TAAGATTA), a location resembling an operator-like element (Fig. 5). The promoters predicted upstream of *orf1* and *orf4* had no obvious regulatory sequences, and the *orf1* promoter was similar to the strong canonical *Escherichia coli* promoter (37). A summary of all relevant properties of the encoded proteins in the bacteriocin gene cluster is presented in Table 2.

**Matrix-assisted laser desorption ionization–time of flight mass spectrometry reveals a bacteriocin peptide in active fractions.** Given that there were several bacteriocin-like peptides that could potentially contribute to antimicrobial activity, we subjected all active fractions from the purification (fractions 11 to 13) to matrix-assisted

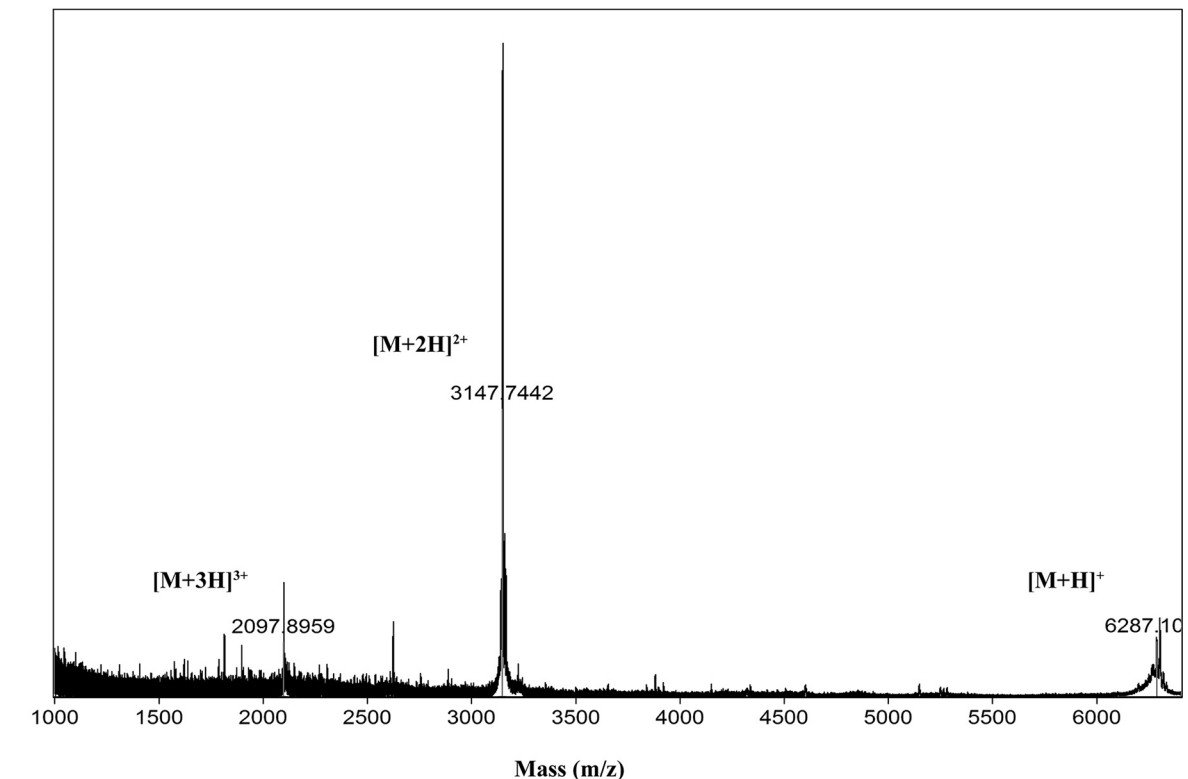

**FIG 6** MALDI-TOF mass analysis of the most active fraction from RPC. The peak labeled at 6,287.10 Da indicates the presence of ubericin K. The peaks at 2,097.90 *m/z* and 3,147.74 *m/z* likely represent the triply and doubly charged ions, respectively.

laser desorption ionization–time of flight mass spectrometry (MALDI-TOF MS) for mass determination. A distinct peak closely corresponding to the mass of the predicted mature peptide of ORF10 could be seen at 6,287.10 *m/z* (Fig. 6); the highest intensity peak at 3,147.74 *m/z* likely represents the doubly charged ion of the same molecule. Similarly, the peak at 2,097.8959 *m/z* is the triply charged ion. The theoretical monoisotopic mass of the mature sequence of ORF10 with oxidized cysteines is 6,290 Da (2,097.89 × 3 − 3 = 6,290.67). Although other masses were also present in purified samples, they did not correspond to the mass of any of the other bacteriocin-like peptides found in the cluster. This result suggests that the predicted mature part of ORF10, hereafter called ubericin K, was primarily responsible for the antimicrobial activity observed.

**In vitro expression of ubericin K, bioactivity, and antimicrobial spectrum.** Previous research has shown that bacteriocins are sometimes incorrectly identified due to the copurification of small and undetectable amounts of other bacteriocin peptides (38). As the producer in our case was predicted to encode numerous bacteriocin-like peptides, we could not exclude the possibility that other antimicrobial peptides in the sample also contributed to the observed antimicrobial activity. To avoid this potential problem, we sought to synthesize the mature peptide of ubericin K by *in vitro* synthesis (IVS) to confirm its bioactivity in the absence of any copurified molecules. As shown in Fig. 7, IVS-ubericin K inhibited the growth of the same indicators used in the initial screening, thus confirming the bioactivity of ubericin K.

As mentioned above, ubericin K shows sequence similarities to lactococcin A, garvieacin Q, and bovicin 255. The antimicrobial activity of lactococcin A appears to be confined to members of the genus *Lactococcus* (12), while garvieacin Q inhibits a broader range of organisms that include *En. faecium*, *La. garvieae*, and *Listeria monocytogenes* (13). Bovicin 255 was also active against a species of *Enterococcus*; however, this bacteriocin was only tested against ruminal bacteria (15). To examine the

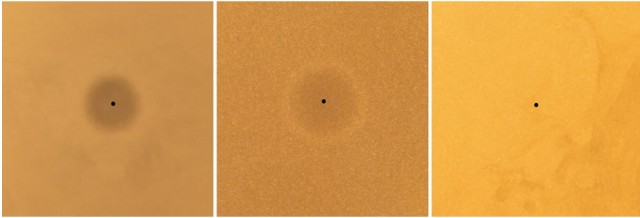

**FIG 7** Reaction mixture (3 μl) from *in vitro*-synthesized ubericin K spotted on a lawn of each of the three indicators used in the screening. *En. faecalis* LMGT 2333 (left), *Str. dysgalactiae* LMGT 3890 (middle), and *Sa. aureus* LMGT 3023 (right).

inhibitory range of ubericin K, a selection of microorganisms was tested for growth inhibition by both *in vitro*-synthesized and reverse-phase column (RPC)-purified ubericin K. As shown in Table 3, the IVS-ubericin K displayed an almost identical inhibition spectrum as the purified ubericin K. Most notable was a very potent antilisterial activity, with strong inhibition also of *En. faecium*, *Str. uberis*, and *La. lactis*. We also noted that resistant colonies were readily visible in the inhibition zones from both IVS-ubericin K and purified ubericin K with *Li. monocytogenes*, *Enterococcus*, and *La. lactis*, a phenomenon also seen with other man-PTS-targeting bacteriocins (data not shown) (24).

**Ubericin K is a pore former targeting the mannose PTS system on sensitive cells.** Lactococcin A and garvieacin Q are both known to use the sugar transporter man-PTS as the receptor to target sensitive cells (17, 39), and we wanted to know whether this was also true for ubericin K. To examine this, we exposed *La. lactis* IL1403 (40) and a mutant where the operon encoding the man-PTS system has been deleted (strain B464) (17) to the *in vitro*-synthesized ubericin K. In this experiment, we also included the RPC-purified ubericin K and nisin A, a lantibiotic with a mechanism of action independent of man-PTS, for comparison. As seen in Fig. 8, both IVS-ubericin K and the purified ubericin K were active against the wild-type strain but not the man-mutant. Nisin A, which uses lipid II as a docking molecule, was active against both strains. These results together provide strong evidence that ubericin K was produced by *Str. uberis* LMGT 4214 and that an intact man-PTS is required for the sensitivity toward this bacteriocin.

**TABLE 3** Inhibition spectrum of RPC-purified ubericin K and *in vitro*-synthesized (IVS) ubericin K in a spot-on-lawn assay

| Indicator strain[a] | Purified ubericin K[b] | IVS-ubericin K[b] |
|---|---|---|
| *Streptococcus dysgalactiae* LMGT 3890 | ++ | + |
| *Streptococcus dysgalactiae* LMGT 3899 | ++ | ++ |
| *Lactococcus lactis* IL1403 | +++ | +++ |
| *Bacillus cereus* LMGT 2805 | − | − |
| *Bacillus cereus* ATCC 9136B | − | − |
| *Enterococcus faecalis* LMGT 2333 | + | + |
| *Enterococcus faecalis* LMGT 3088 | + | + |
| *Enterococcus faecium* LMGT 2763 | +++ | +++ |
| *Enterococcus faecium* LMGT 2772 | +++ | +++ |
| *Lactococcus lactis* IL1403 | +++ | +++ |
| *Lactobacillus curvatus* LMGT 2353 | ++ | ++ |
| *Lactobacillus garvieae* LMGT 3390 | ++ | ++ |
| *Listeria monocytogenes* LMGT 2651 | +++ | +++ |
| *Listeria monocytogenes* LMGT 2604 | +++ | ++ |
| *Listeria innocua* LMGT 2785 | +++ | +++ |
| *Staphylococcus aureus* LMGT 3023 | − | − |
| *Staphylococcus aureus* LMGT 3242 | − | − |
| *Streptococcus uberis* LMGT 3912 | +++ | +++ |
| *Escherichia coli* TG1 | − | − |

[a]LMGT, Laboratory of Microbial Gene Technology, Norwegian University of Life Sciences, Ås, Norway.
[b]+++, clear zone of inhibition; ++, smaller clear zone of inhibition; +, visible/diffuse inhibition; −, no inhibition.

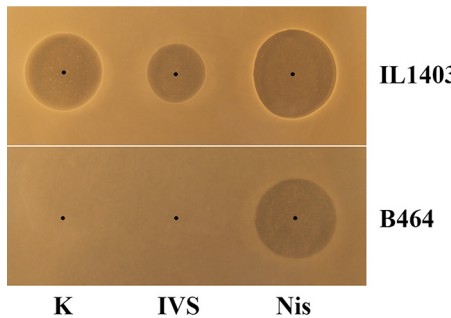

**FIG 8** Inhibition of wild-type *Lactococcus lactis* IL1403 and a *ptn* (man-PTS) deletion mutant of IL1403 (B464) by *in vitro*-synthesized ubericin K (IVS), RPC-purified active fraction of ubericin K (K), and nisin A (Nis). No inhibition is observed by ubericin K with the mutant. Nisin A was used as a positive control.

We have previously constructed a sensor strain of *Li. monocytogenes* expressing pHlourin (41). pHluorin is a pH-sensitive green fluorescent protein that has different fluorescence emission spectra dependent on the local pH (42). Thus, by keeping the intra- and extracellular pH at different values, one can observe a shift in emission pattern if there is a leakage of protons across the membrane (41). The activity of ubericin K toward *Li. monocytogenes* that we observed previously allowed us to use this sensor strain to investigate whether ubericin K kills cells by forming pores on target cells. When the sensor strain was exposed to nisin A, which is a known pore former, a significant reduction in the fluorescence emission at 510 nm was seen (excitation at 400 nm over 470 nm) as expected (Fig. 9). The same was seen with the detergent cetyltrimethylammonium bromide (CTAB), which is commonly used as a positive control for a membrane disruption agent. Interestingly, ubericin K also caused a similar shift in emission, thus indicating that this bacteriocin has a mode of action that involves pore formation. Thus, ubericin K kills sensitive cells by man-PTS-dependent pore formation. Pore formation was also evident for pediocin PA-1, a class IIa bacteriocin targeting man-PTS (43).

## DISCUSSION

In this study, we screened milk samples from dairy herds in Norway and succeeded in isolating a strain of *Str. uberis* that produces a new bacteriocin that kills closely related species as well as *En. faecium* and *Listeria* spp. *Str. uberis* is an organism frequently detected

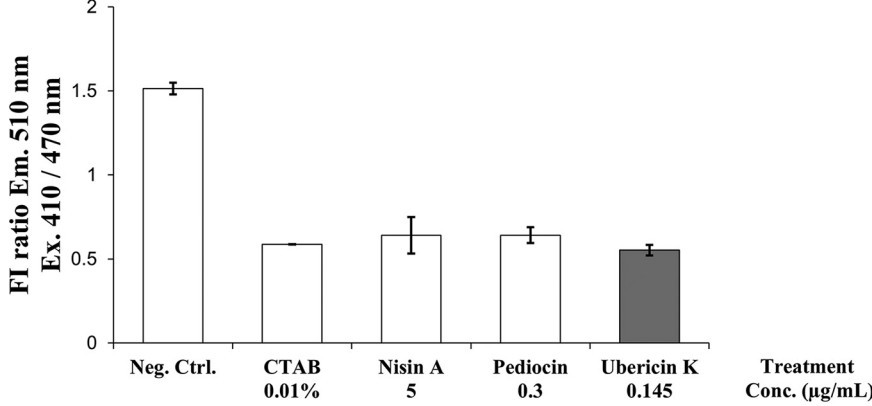

**FIG 9** Assay for measuring a drop in intracellular pH during exposure to antimicrobials. A ratiometric pH-sensitive variant of green fluorescent protein (GFP) is expressed in *Li. monocytogenes* EGDe/pNZ-P$_{help}$-pHluorin, and a lower ratio of fluorescence from excitation at 410 nm over 470 nm indicates reduced pH. Ubericin K was equally as effective as the positive control CTAB (0.01%) in causing a drop in intracellular pH.

from the milk and udders of dairy cows and is also recognized as one of the primary causative agents of mastitis in Norway and worldwide (44–47). Virulence determinants required for host invasion and colonization of *Str. uberis* have not been clearly defined and likely involve more complex population dynamics and interaction (48). A key component in population dynamics of streptococci is the intercommunication of strains with peptide pheromones and the intra- and interspecies competition by the production of bacteriocins (49). *Str. uberis* LMGT 4214 shows strong antagonism toward other mastitis-associated strains and could therefore potentially outcompete virulent strains in the udder. However, the pathogenic potential of *Str. uberis* LMGT 4214 itself has not yet been established. Optionally, purified bacteriocin from the producer could have potential in treatment and prevention of mastitis, as has been demonstrated previously for lacticin 3147 and micrococcin P1 (50, 51).

Antimicrobial activity was purified from the culture supernatant by methods commonly used for bacteriocins based on their general physicochemical characteristics, including small size and cationic and hydrophobic nature. Genomic analysis of *Str. uberis* LMGT 4214 revealed a multibacteriocin gene cluster potentially expressing three novel bacteriocins in addition to a three-component regulatory system (composed of an HPK, an RR, and a peptide pheromone) and a dedicated ABC transporter. However, when analyzing the active fractions from RP-FPLC, only one peak showed antimicrobial activity.

The three-component system suggested that bacteriocin production is regulated by a quorum-sensing mechanism, as has previously been described in other Gram-positive bacteria, such as for the plantaricins EF and JK, sakacin A, and the Blp bacteriocins in *Lactobacillus plantarum*, *Lactobacillus sakei*, and *Streptococcus pneumoniae*, respectively (38, 52, 53). In this system, an activated response regulator binds to conserved direct repeats at regulated promoters as a homodimer (54, 55). In the bacteriocin cluster of *Str. uberis* LMGT 4214, two putative promoters were found with proximal direct repeats, suggesting the presence of at least two regulated promoters. The arrangement of the direct repeats is such that they face toward the same side of the DNA, thereby facilitating a dimeric binding of a regulator, a function that is likely performed by the response regulator encoded by *orf1*. It is feasible that a dimeric regulator bound to the direct repeats can assist RNA polymerase in binding to its promoter located only 35 nt downstream. Another interesting feature identified is the second pair of direct repeats, which are partially overlapping the predicted −10 box of the two predicted regulated promoters. In view of its location, this feature resembles an operator sequence for which a repressor could bind and sterically block access to the promoter. Regulated promoters involving both activating regulatory elements and operator are known in nature. An example of this is the classical *lac* operon, which involves the catabolite gene activator protein (CAP) site and the *lac* operator in *Es. coli* (56). However, such a regulation has, to our knowledge, not yet been found for any bacteriocins, and whether the repeats identified in the bacteriocin cluster of *Str. uberis* LMGT 4214 serve such functions requires further investigation in future work, as this aspect is beyond the scope of the present study.

Of the bacteriocins found in the cluster, two of the bacteriocins belong to the class IId family and one to the class IIb family, all being followed by a gene or genes encoding hydrophobic proteins, which likely play a role in immunity. The predicted bacteriocin-like peptides is of a relatively small size (<7 kDa) with a high pI and an N-terminal 15- to 30-amino acid leader sequence with a GG-type cleavage motif. Maturation and export of bacteriocins with this type of leader sequence normally use a dedicated ABC transporter and peptidase where maturation occurs concomitant with export, a function likely executed by the encoded ABC transporter in the locus. Nevertheless, calculating the expected mass of the four bacteriocin-like peptides found in the bacteriocin gene cluster only provided one match to the mass spectrometry peak at approximately 6,290 Da (*m/z*), which is the mass of mature ubericin K. To ensure that the antimicrobial activity measured from the purified sample was due to ubericin K and not to unknown antimicrobial contaminants, the peptide was synthesized *in vitro*. This showed that IVS-ubericin K has an identical inhibition spectrum as purified ubericin K and that both required an intact man-PTS for antimicrobial activity, thus confirming that ubericin K

alone could be responsible for the antimicrobial activity. Resistant colonies in the inhibition zones from both IVS-ubericin K and purified ubericin K indicates that resistance to the bacteriocin is likely prevalent in nature; this is a challenge that must be addressed for the clinical application of this class of antimicrobials.

Ubericin K was *in vitro* synthesized as the mature peptide but with an added initiator methionine (*N*-formylmethionine) that is not present in the native mature bacteriocin. It is worth mentioning that IVS-ubericin K activity was the highest immediately after synthesis. Activity of the reaction mixture was significantly reduced following storage, showing a reduction in the diameter of inhibition zones by about 4-fold after 24 h at room temperature and only a faint zone of inhibition following overnight storage at 4°C or a freeze-thaw cycle (data not shown). A possible explanation for this loss could be the formation of multimeric complexes or aggregates due to the cationic and hydrophobic properties and low solubility under the basic aqueous conditions required for *in vitro* synthesis. The formation of inactive precipitates following storage at 4°C has been documented for lactococcin A (12). Such aggregates would also be expected to form in the supernatant and aqueous buffers during purification. The apparent increase in the amount of bacteriocin (approximately 200% yield) following reverse-phase purification is then probably from the dissociation and resolubilization of aggregates by the isopropyl alcohol/trifluoroacetic acid (TFA). The loss of activity could also be a result of, or exacerbated by, oxidation of the added initiator methionine, a phenomenon that also has been observed with pediocin PA-1 and lactococcin B (57, 58). In addition, attempts at purifying *in vitro*-synthesized ubericin K from the reaction mixture using 100,000-molecular weight cutoff filters to remove the macromolecules necessary for the reaction, as recommended by the manufacturer, were not successful, as activity was lost in the filtrate. We did not ascertain if this was due to aggregates or from the adsorption of ubericin K to the filter material. Thus, to fully take advantage of *in vitro* synthesis to characterize bacteriocins mined from genomic databases, this instability needs to be solved in future research.

Ubericin K was shown to disturb the pH homeostasis of sensitive cells in the same manner as the pore-forming bacteriocin nisin A and the potent antilisterial bacteriocin pediocin PA-1. Pore formation in target cells by the pediocin-like class IIa has been well established (59). In addition, the one-peptide nonpediocin-like class IId bacteriocin lactococcin A also causes pore formation and loss of the proton motive force (21). The pore-forming ability of garvieacin Q and bovicin 255 has not been established, but the results presented here strongly suggest a similar mechanism based on their sequence homology.

Despite having a multibacteriocin gene cluster, the presence of other bacteriocins than ubericin K was not apparent. It is possible that the other putative bacteriocin-like peptides (ORF6, ORF7, and ORF13) have no or low activity against the chosen indicator or they are differently regulated. For ORF6 and ORF7 that likely constitute a two-peptide bacteriocin, no activity would be expected if the peptides were separated into different fractions during purification. This notion is in fact relatively common, as we have previously encountered similar problems during the purification of the multipeptide plantaricins (38) and garvicin KS (29). The regulation, inhibitory spectrum, and activity of these bacteriocin-like peptides thus remain to be determined, and further characterization of this bacteriocin cluster by heterologous expression and *in vitro* synthesis together will help answer those questions.

Genomes uploaded to public databases often contain bacteriocin clusters, and programs, such as BAGEL4 and other annotation software, are continuously getting better at correctly identifying bacteriocin-like peptides and bacteriocin biosynthetic proteins. Indeed, BLAST searches of all bacteriocin-like peptides in the cluster had significant matches to protein sequences annotated as bacteriocins. As such, using data mining to find new bacteriocin-like genes is one possible approach (60–62). However, the major bottleneck in discovering novel bacteriocins with desired properties is their characterization experimentally. *In vitro* protein synthesis is a promising rapid method for

**TABLE 4** Microorganisms used in this study with relevant characteristics

| Bacterial strain[a] | Relevant characteristics | Reference[a] |
|---|---|---|
| *Streptococcus uberis* LMGT 4214 | Bacteriocin producer | This study |
| *Lactococcus lactis* IL1403 | Indicator strain | 40 |
| B464 | *ptn* deletion mutant of IL1403 | 17 |
| *Escherichia coli* DH5α | Cloning and plasmid propagation host | Invitrogen (Cat. No. 18265-017) |
| *Listeria monocytogenes* EGDe/pNZ-P<sub>help</sub>-pHluorin | A clone expressing a pHluorin protein used to measure pore formation ability of bacteriocins | 41 |
| *Enterococcus faecalis* LMGT 2333 | Indicator strain used in the screening | Lab collection (LMGT), Norway |
| *Staphylococcus aureus* LMGT 3023 | Indicator strain used in the screening | Lab collection (LMGT), Norway |
| *Streptococcus dysgalactiae* LMGT 3890 | Indicator strain used in the screening | Lab collection (LMGT), Norway |
| *Streptococcus uberis* LMGT 3912 | Indicator strain used to monitor activity for bacteriocin purification. Mastitis pathogen. | Lab collection (LMGT), Norway |

[a]LMGT, Laboratory of Microbial Gene Technology, Norwegian University of Life Sciences, Ås, Norway.

screening and testing new bacteriocins, as the small unmodified and leaderless bacteriocins are largely unstructured in aqueous solution. This method allows for a streamlined pipeline for characterizing bacteriocins from sequence data (63). Further characterization of the many bacteriocins from streptococci, among others, will be valuable for devising alternative strategies for the treatment and prevention of mastitis and other infections as well as to understand the interstrain competition in these environments.

## MATERIALS AND METHODS

**Strains and growth conditions.** Bacterial strains used in this study are listed in Tables 3 and 4. All bacterial strains in Table 4 were grown in brain heart infusion (BHI) (Oxoid) at 37°C, except for strains of *Lactococcus lactis*, which were propagated in M17 broth (Oxoid) supplemented with 0.5% (wt/vol) glucose (GM17) at 30°C, and *Escherichia coli* NEB 5-alpha (New England BioLabs), which was grown in LB (Oxoid) containing 100 µg/ml ampicillin at 37°C with shaking. All strains used for the determination of the antimicrobial spectrum (Table 3) were grown in BHI at 30°C.

**Bacteriocin screening.** Bovine raw milk samples were collected from individual cows selected from two dairy herds. Sample collection as well as the microbiota content of the samples have been described previously by Porcellato et al. (25). Screening of bacteriocin producers was performed using a multisoft agar overlay method as follows. Samples of raw bovine milk were first diluted in saline (0.9% NaCl) to ensure a good distribution of colonies (50 to 500 colonies/plate) before being mixed with soft agar and poured onto BHI agar. A second layer of BHI soft agar was poured on top before the plates were incubated at 30°C overnight. On the following day, an overnight culture of the indicator strain (*Enterococcus faecalis* LMGT 2333, *Staphylococcus aureus* LMGT 3023, or *Streptococcus dysgalactiae* LMGT 3890) was diluted 50-fold in BHI soft agar and poured evenly as a top layer. Following incubation at 37°C overnight, the plates were inspected for zones of growth inhibition of the indicator. The colonies producing inhibition zones were restreaked on new BHI plates to obtain pure cultures, and bacteriocin production was confirmed by a second inhibition test before frozen cultures in 15% glycerol were made and stored at −80°C until use.

**DNA extraction, repetitive element PCR fingerprinting and 16S rRNA gene sequencing.** Genomic DNA isolation and purification was performed using a GenElute bacterial genomic DNA kit (Sigma-Aldrich). Isolated DNA was used as a template for both repetitive element PCR (rep-PCR) and 16S rRNA gene amplification. rep-PCR was performed using the primer pair ERIC1R and ERIC2 (Table 5), as described by Versalovic et al. (64). The 16S rRNA gene was amplified using the primers 16S-11F and 16S-12R (Table 5), and the resulting PCR product was sequenced using the same primers by Sanger sequencing (Eurofins Genomics). A contig of the resulting reads was constructed using CAP3 (65) and searched against the NCBI rRNA/ITS database.

**Bacteriocin purification.** The bacteriocin was purified from the supernatant of 1 liter of overnight culture. Cells were removed by centrifugation at 10,000 × *g* for 30 min, and the bacteriocin in the supernatant was precipitated by the addition of ammonium sulfate (60% saturation, 4°C). After centrifugation for 40 min at 12,000 × *g*, the bacteriocin precipitate was dissolved in Milli-Q water (Merck Millipore) and adjusted to a pH of 4 by the addition of 1 M hydrochloric acid and subjected to cation-exchange chromatography using a HIPrep 16/10 SP-XL column (GE Healthcare Biosciences). The column was washed with 5 column volumes (CV) of 20 mM sodium phosphate buffer at a pH of 6.8 before the bacteriocin was eluted from the column with 5 CV of 1 M sodium chloride (unbuffered). The eluate containing the bacteriocin was applied on a Resource reverse-phase chromatography (RPC) column (1 ml) (GE Healthcare Biosciences) connected to an ÄKTA purifier system (Amersham Pharmacia Biotech). The column was equilibrated with 20 CV of 0.1% TFA before loading the sample and eluted with a linear gradient of 15% to 60% isopropyl alcohol (Merck) containing 0.15% (vol/vol) TFA at a rate of 1 ml/min. The concentration of ubericin K in the final RPC-purified fraction was estimated using the Qubit protein assay kit (Invitrogen). Activity from each step of the purification procedure was assessed using the indicator strain *Str. uberis* LMGT 3912, which was isolated from a case of clinical mastitis.

**TABLE 5** Oligonucleotides used in this study

| Oligonucleotide | Sequence (5′ to 3′) |
| --- | --- |
| 16S-11F | TAACACATGCAAGTCGAACG |
| 16S-12R | AGGGTTGCGCTCGTT |
| ERIC1R | ATGTAAGCTCCTGGGGATTCAC |
| ERIC2 | AAGTAAGTGACTGGGGTGAGCG |
| KC1F | CGTAATTCATATGGCTAAAGGTGTCTGTAAGTATG |
| KC1R | ATGGATCCGTTTACCCTCTATTTCCGTGG |
| KIF | TGCTAGCCCCGCGAAATTAATACG |
| KGAP1F | ACATCGACTTATCTTGCACG |
| KGAP2R | CATACAACTCTTCAACATGTCG |

**Bacteriocin assays.** Antimicrobial activity in solutions obtained from each step in the purification procedure was determined using a microtiter plate assay (12). Twofold dilutions of sample in BHI were prepared in microtiter plates to a volume of 100 $\mu$l per well. Each well was then inoculated with 100 $\mu$l of a 25-fold diluted overnight culture of the indicator *Str. uberis* LMGT 3912 (50-fold final dilution). After incubation at 37°C for approximately 8 h, the turbidity was measured spectrophotometrically at 600 nm using a SPECTROstar Nano reader (BMG Labtech). One bacteriocin unit (BU) was defined as the minimum amount of the antimicrobial that inhibited growth of the indicator strain by at least 50% in 200 $\mu$l of culture.

The inhibition spectrum of RPC-purified ubericin K (most active fraction) and *in vitro*-synthesized ubericin K was performed as a spot-on-lawn assay. Indicator strains were grown overnight in BHI at 30°C and then diluted 50-fold in 5 ml of BHI soft agar (0.8% agarose) and poured over a base layer of BHI agar (1.5% agarose). After solidification, 3 $\mu$l of purified antimicrobial or *in vitro*-synthesized ubericin K was spotted on the plates. Plates were inspected visually for inhibition zones after overnight incubation at 30°C. Nisin A (N5764, Sigma-Aldrich) was included as a comparison and prepared with a potency of ≥40,000 IU/ml in 0.05% (vol/vol) acetic acid, insolubles were removed by centrifugation, and remaining nisin A solution was sterile filtered (0.22-$\mu$m pore size; Millipore).

Pore formation in target cell membranes was tested using the recently published fluorescent reporter strain *Li. monocytogenes* EGDe/pNZ-P$_{help}$-pHluorin (41). This strain expresses the fluorescent protein pHluorin that has a bimodal excitation spectrum showing ratiometric pH-dependent changes in fluorescence intensity (42). In *Listeria* minimal buffer (LMB) at pH 6.5, untreated *Li. monocytogenes* EGDe/pNZ-P$_{help}$-pHluorin cells are able to maintain intracellular pH. However, in the presence of membrane-damaging compounds, the intracellular pH rapidly drops, resulting in a characteristic change in fluorescence at the two excitation peaks. For assays, an overnight culture of *Li. monocytogenes* EGDe/pNZ-P$_{help}$-pHluorin was harvested by centrifugation (3,000 × *g* for 10 min at 4°C), washed once in phosphate-buffered saline, and resuspended in LMB at pH 6.5 (41) to an optical density at 600 nm (OD$_{600}$) of 3. One hundred microliters of this suspension was added to individual wells of the 96-well screening plates. Then, 100 $\mu$l of a sample was added and plates were vortexed for 10 s, wrapped in aluminum foil, and incubated for 1 h at room temperature in the dark. Fluorescence was measured using a Tecan Infinite M200 microplate reader with excitation at 400/9 and 470/9 nm and emission at 510/20 nm. The ratios of emission intensities after excitation at 400 and 470 nm were calculated.

**MALDI-TOF mass spectrometry.** Acquisition of mass spectrometry data was performed on an UltrafleXtreme III TOF/TOF (Bruker Daltonics) MALDI-TOF mass spectrometer operated in reflectron mode. The instrument was set to analyze positively charged ions in the range of 1,400 to 6,600 *m/z* and had been externally calibrated in the *m/z* range of 700 to 3,100 using the peptide calibration standard II (Bruker Daltonics). The RPC-purified active fraction was mixed 1:1 with matrix solution ($\alpha$-cyano-4-hydroxycinnamic acid [HCCA]) as recommended by the supplier (Bruker Daltonics) and applied to a stainless steel MALDI target plate (Bruker Daltonics).

**Whole-genome sequencing.** Genomic DNA isolation and purification was performed using a GenElute bacterial genomic DNA kit (Sigma-Aldrich). The sequencing libraries were prepared using a Nextera XT DNA sample prep kit (Illumina, San Diego, CA, USA) according to the manufacturer's instructions. Sequencing was performed using the Illumina MiSeq platform (Illumina) and V3 chemistry. Reads were error corrected and assembled *de novo* using SPAdes v3.14.1 (66). The obtained contigs were annotated by InterProScan (67) as well as submitted to the BAGEL4 web server to search for potential bacteriocin gene clusters (26). Initial assembly did not result in a complete cluster due to a lack of coverage upstream of ORF2. Therefore, primers KGAP1F and KGAP2R (Table 5) were designed based on the initial assembly and used to fill the gap region. The two primers were used in PCR, and the product was sequenced by Sanger sequencing (Eurofins Genomics).

***In vitro* protein synthesis.** The mature bacteriocin peptide sequence was synthesized *in vitro* using the PURExpress *in vitro* protein synthesis kit (New England BioLabs). First, the DNA sequence encoding the mature peptide was amplified from the producer using the primer pair KC1F and KC1R containing a start codon (ATG) and restriction sites NdeI and BamHI. All primers used in his study are listed in Table 5. The amplified product was purified with Macherey-Nagel PCR cleanup and gel extraction kit (Macherey-Nagel). The resulting amplicon and the DHFR control plasmid supplied with the PURExpress kit were digested with NdeI and BamHI (Thermo Fisher Scientific) according to manufacturer's recommendations. Digests were mixed in a molar ratio of 3:1 (insert to vector) and ligated with T4 DNA ligase (New England BioLabs) at room temperature for 10 min, and the ligation mixture was cloned into competent *Es. coli* NEB 5-alpha (C2987) cells following the high efficiency transformation protocol supplied by the manufacturer. The construct was isolated with an EZNA plasmid minikit I (Omega Bio-Tek) and verified by sequencing using the K1F primer before *in vitro* synthesis. Approximately 280 ng of plasmid, as estimated by a NanoDrop 2000 spectrophotometer (Thermo Fischer Scientific), was used

as the template for *in vitro* protein synthesis. The reaction mixture was incubated at 37°C for 4 h, diluted 2-fold with 20 mM magnesium acetate (Sigma-Aldrich), and used immediately without further purification.

**Data availability.** The entire bacteriocin gene cluster has been deposited in GenBank under accession number MZ189362.

## ACKNOWLEDGMENTS

The work has been funded by the Research Council of Norway (project number 275190) and by the Norway Grants 2014-2021 via the National Centre for Research and Development (grant number NOR/POLNOR/PrevEco/0021/2019-00), by the Norwegian Foundation for Research Levy on Agricultural Products (FFL), and the Norwegian Agricultural Agreement Research Fund (JA) (grant number 267623). M.K. is supported by a JPIAMR grant from the Research Council of Norway (project number 296906).

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
