## [Reviewer comments · Microbiology Spectrum]

**Microbiology
Spectrum**

Ubericin K, a new pore-forming bacteriocin targeting mannose-PTS

Thomas Oftedal, Kirill Ovchinnikov, Kai Hestad, Oliver Goldbeck, Davide Porcellato, Judith Narvhus, Christian Riedel, Morten Kjos, and Dzung Diep

Corresponding Author(s): Dzung Diep, Norwegian University of Life Sciences

Review Timeline:

Submission Date:	May 14, 2021
Editorial Decision:	July 5, 2021
Revision Received:	August 26, 2021
Accepted:	September 10, 2021

Editor: Kate Howell

Reviewer(s): The reviewers have opted to remain anonymous.

Transaction Report:

DOI: <https://doi.org/10.1128/Spectrum.00299-21>

July 4, 2021

Prof. Dzung B Diep
Norwegian University of Life Sciences
Laboratory of Microbial Gene Technology
Dept. of Chemistry, Biotechnology & Food Science (IKBM)
P.O. Box 5003
1432 Aas
Norway

Re: Spectrum00299-21 (Ubericin K, a new pore-forming bacteriocin targeting mannose-PTS)

Dear Prof. Dzung B Diep:

The reviewers were very positive about your paper and findings, but both have raised questions about the purification and verification of your bacteriocin. I ask you to carefully consider these suggestions.

Thank you for submitting your manuscript to Microbiology Spectrum. When submitting the revised version of your paper, please provide (1) point-by-point responses to the issues raised by the reviewers as file type "Response to Reviewers," not in your cover letter, and (2) a PDF file that indicates the changes from the original submission (by highlighting or underlining the changes) as file type "Marked Up Manuscript - For Review Only". Please use this link to submit your revised manuscript - we strongly recommend that you submit your paper within the next 60 days or reach out to me. Detailed information on submitting your revised paper are below.

Link Not Available

Sincerely,

Kate Howell

Journals Department
Reviewer comments:

Reviewer #1 (Comments for the Author):

The manuscript describes the isolation of an antimicrobial producing strain from Norwegian bovine raw milk that displays activity against mastitis pathogens. 16s sequencing identified the strain as *Streptococcus uberis* while whole genome sequencing revealed the presence of a multi bacteriocin gene cluster encoding one Class IIb bacteriocin, two class IIc bacteriocins, a three-component regulatory system and a dedicated ABC transporter. Attempts to purify the antimicrobials produced by the strain resulted in an active RP-HPLC fraction that contained a mass correlating with Ubericin K, one of the Class IIc bacteriocins encoded on the gene cluster. Ubericin K was synthesised and shown to target Mannose PTS.

While characterisation of the bacteriocin cluster and the potentially novel bacteriocins encoded by it is of interest the presence of a multi-bacteriocin gene cluster requires extensive characterisation of the strain to establish how many of the bacteriocins are being produced and in what quantity. As the bacteriocin sequences are available MALDI TOF MS can be used to assess eluents at each purification stage for putative bacteriocin masses. The assessment of bacteriocin production by the strain requires further work. Specifically, bacteriocin production by the cells needs to be assessed particularly as there is such low activity (20 BU/ml) associated with the culture cell free supernatant. In addition, further attempts should be made to determine if the 2 component bacteriocin is being produced. The manuscript correctly states that individual peptides may be inactive but it is not clear if any attempts were made to find the two component bacteriocin. The use of MALDI TOF mass spectrometry should help to assess this. Ammonium sulphate precipitation may not be the best starting point for purification and the use of cation exchange and/or hydrophobic interaction resins together with MALDI TOF analysis of active eluents may be a better starting point. It might also be worth considering how production could be improved (different media, growth conditions) as starting with a higher BU/ml will increase the chances of obtaining pure peptide(s).

Specific comments

Line 38 Which novel method was described to rapidly characterise bacteriocins from complex samples?

Line 60 Are circular bacteriocins considered Class I or Class II?

Line 140 states that bacteriocin was eluted in 1M sodium chloride. Is the sodium chloride in 20 mM sodium phosphate buffer?

Line 151 Why wasn't activity assessed on solid agar plates to assess eluents from each purification step? This would give a much clearer indication of the effectiveness of the different purification steps. Were any of the active eluents from the purification protocol analysed by MALDI TOF MS?

Line 153 Is *S. parauberis* a mastitis pathogen?

Line 163 state the % nisin in the Sigma nisin - is it 2.5%? Following resuspension and purification what is the final concentration of nisin assayed?

Line 187 The peptide calibration mix is out range (700-3100 Da) for Ubericin K (6294.85 Da)

Line 252-255 This statement is misleading. Fractions 21-23 elute in a peak with an apex at 22 minutes. The chromatography could be improved by a) running the sample on a shallower gradient (15-40% perhaps) and b) extending the run time to enable better separation.

Line 255 "a rather complex nature of the antimicrobial activity" This is incorrect as the nature of the antimicrobial activity has no bearing on purification.

Line 258 "Given the elusive nature of the antimicrobial activity" What does this mean? If the antimicrobial activity is elusive is the strain suitable as a mastitis treatment?

Line 329-331 Strong evidence that ubericin K is responsible for the antimicrobial activity requires an

active HPLC fraction and a mass spectrum showing a single mass correlating to the ubericin K mass. This information should be shown in figure 2. As the gene cluster also contains a two component bacteriocin and another Class IId bacteriocin the efforts made to show that they are not being produced should be described. As the putative bacteriocin sequences are available the purification protocol can be customised to purify the specific bacteriocins.

Line 334 "the nature of the bacteriocins is sometimes incorrectly identified" - what does this mean exactly?

Lines 400-405 The conclusions drawn here are very speculative. The starting bacteriocin activity was very, very low (20 BU/ml) suggesting a very low yield in a complex background (BHI broth). The impurities are most likely media contaminants and the purification procedure requires optimisation to remove these. There is no connection between this and the fact that the strain contains a multi-bacteriocin gene cluster.

Line 411-416 This is repetition of the results lines 306-320

Line 442 Activity from the synthetic peptide confirms that ubericin K is active and likely to be produced by the strain. It does not confirm that ubericin K is the sole antimicrobial produced by the strain.

Lines 446-465 Loss of activity in the synthetic peptide was attributed to hydrophobicity and aggregate formation. Ubericin K elutes at ~ 30% acetonitrile on the HPLC gradient suggesting that the bacteriocin is not very hydrophobic and is likely to be soluble.

How does aggregation result in increased activity?

What evidence is there for aggregation - was precipitation observed? Was MALDI TOF MS used to detect the aggregates?

It is not stated say how the 100 KDa cut off membrane was expected to help with purification. Did the activity pass through the membrane or was it retained by the membrane?

Bacteriocins are very soluble in Isopropyl alcohol /TFA - what evidence is there that it is associated with dissociation and re-solubilisation of bacteriocins of similar hydrophobicity?

Is there evidence that formylmethionine oxidises as readily as methionine?

Line 473-474 Further efforts should be made to show that the two component bacteriocin is not being produced by the strain. As stated in the manuscript the individual peptides can be inactive therefore assaying individual fractions with the other fractions is required to check for the two component bacteriocin. Alternatively, all fractions should be analysed by MALDI TOF MS for the putative bacteriocin masses.

Figures and tables

Figure 1 Attempts should be made to assess which bacteriocins are associated with the cell surface

Figure 2 and Figure 6 should be combined and indicator plates showing the zones of inhibition at each purification stage should be shown ie starting zone from culture broth, zone from Ammonium sulphate precipitate, cation exchange and the HPLC fraction. Alternatively, the zones could be shown in Table 3.

Figure 4 Include theoretical masses of the mature putative bacteriocins

Suggestions

Line 21 change "revealed" to "resulted in the detection of" as sentence is poorly phrased and misleading.

Line 22 MALDI TOF mass spectrometry - insert TOF

Line 27 "a" target receptor

Line 37 We also describe ... rather than explain

Line 46 Is spp. necessary after Enterococcus? Remove italics from spp

Line 73, lactococcin A, bacteriocin SJ and garviecin Q are mentioned together more than once in the manuscript. Consider keeping word order the same throughout the manuscript for clarity.

Line 84 active at picomolar concentrations

Line 102 which was grown in
Line 103 determination of the antimicrobial spectrum
Line 110 Samples of raw bovine milk were first diluted in
Line 112 soft agar - changed to softagar further on - keep consistent throughout manuscript - two words are better
Line 114 aureus should be in italics
Line 123 using a GenElute....
Line 139 Is it sodium phosphate buffer? Please specify
Line 141 applied to a Reversed ...
Line 150 purification procedure was
Line 167 delete "by"
Line 170 Capital letters for Minimal and Buffer
Line 183 MALDI TOF mass spectrometry
Line 185 include TOF (Time of Flight)
Line 217 10 minutes and the ligation mixture cloned...
Line 218 protocol supplied by the manufacturer
Line 228 S. should be in italics
Line 230 a high abundance
Line 237 enterolysin A gene (33) that encode enterolysin A, a well characterised..
Line 239 delete "appeared to harbour" It's there so suggest "encoded" or "contained an uncharacterised"
Line 240 delete "with regard to nature of antimicrobials" as it doesn't make sense. Potential novelty?
Line 259 was used to identify
Line 260 combined with an in silico
Line 267 calculated size - do you mean peptide length (number of amino acids)?
Line 272 Format has changed and this paragraph is tabbed as are many other paragraphs going forward
Line 279 Mature bacteriocin sequence or something similar rather than "part of"
Line 281 Delete "itself"
Line 291 as a query
Line 292 a similar gene cluster to LMTG 4214
Line 306 An in silico search
Line 322 MALDI TOF mass spectrometry
Line 325 MALDI TOF mass spectrometry
Line 328 Other masses were frequently detected
Line 389 "have" rather than "has"
Line 393 delete "of"
Line 396 "has" rather than "have"

Reviewer #2 (Public repository details (Required)):

The authors have included in the manuscript the Genbank accession number for their genome sequence.

Reviewer #2 (Comments for the Author):

This paper described the identification of a novel bacteriocin, ubericin K, isolated from

Streptococcus uberis. Ubericin K is an unmodified peptide that belongs to class II bacteriocins, is produced with a double-glycine leader peptide, forms pores in sensitive cells and targets the mannose-PTS. The work is very nicely done and the paper is well written.

Some (minor) comments:

- On page 4, the authors describe the classification of bacteriocins in Gram-positive bacteria. The authors should be aware that the classification they describe is based on a review containing an outdated classification. Not only lantibiotics, but also other post-translationally modified bacteriocins, such as sactibiotics, lasso peptides and thiopeptides, are considered to be class I bacteriocins. Cyclic bacteriocins are now part of class I bacteriocins as well. Several reviews with an updated bacteriocin classification have been published in recent years. Please update your classification and reference used.
- On page 8 the authors describe bacteriocin purification. Please add which indicator organism was used to monitor the purification of the peptide.
- Page 12: did the authors use crude expressed peptide samples for their experiments? Or was the expressed peptide further purified? If so, please mention this.
- Line 270: please change "most bacteriocins" into "most unmodified bacteriocins".
- Page 18/Figure 7/Table 5: it is not clear how much material was used for these inhibitory experiments. There is no mentioning of amount, specific activity in AU, or amount in mg protein/ml. The authors should really address this issue.
- Table 5: again no indication of how much ubericin K and IVS ubericin K was used for these inhibitory experiments.

Staff Comments:

Preparing Revision Guidelines

For complete guidelines on revision requirements, please see the Instructions to Authors at [link to page]. **Submissions of a paper that does not conform to Microbiology Spectrum guidelines will delay acceptance of your manuscript.**

Please return the manuscript within 60 days; if you cannot complete the modification within this time period, please contact me. If you do not wish to modify the manuscript and prefer to submit it to another journal, please notify me of your decision immediately so that the manuscript may be formally withdrawn from consideration by Microbiology Spectrum.

If you would like to submit an image for consideration as the Featured Image for an issue, please contact Spectrum staff.

September 10, 2021

Prof. Dzung B Diep
Norwegian University of Life Sciences
Laboratory of Microbial Gene Technology
Dept. of Chemistry, Biotechnology & Food Science (IKBM)
P.O. Box 5003
1432 Aas
Norway

Re: Spectrum00299-21R1 (Ubericin K, a new pore-forming bacteriocin targeting mannose-PTS)

Dear Prof. Dzung B Diep:

Thank you for considering the reviewers comments so carefully and for repeating the purification to answer one of the questions. The manuscript is much improved.

Your manuscript has been accepted, and I am forwarding it to the ASM Journals Department for publication. You will be notified when your proofs are ready to be viewed.

Sincerely,

Kate Howell
Editor, Microbiology Spectrum
